# VoiceLoop: Voice Fitting and Synthesis via a Phonological Loop

**Yaniv Taigman, Lior Wolf, Adam Polyak and Eliya Nachmani**
Facebook AI Research
{yaniv, wolf, adampolyak, eliyan}@fb.com

## Abstract

We present a new neural text to speech (TTS) method that is able to transform text to speech in voices that are sampled in the wild. Unlike other systems, our solution is able to deal with unconstrained voice samples and without requiring aligned phonemes or linguistic features. The network architecture is simpler than those in the existing literature and is based on a novel shifting buffer working memory. The same buffer is used for estimating the attention, computing the output audio, and for updating the buffer itself. The input sentence is encoded using a context-free lookup table that contains one entry per character or phoneme. The speakers are similarly represented by a short vector that can also be fitted to new identities, even with only a few samples. Variability in the generated speech is achieved by priming the buffer prior to generating the audio. Experimental results on several datasets demonstrate convincing capabilities, making TTS accessible to a wider range of applications. In order to promote reproducibility, we release our source code and models[1].

## 1 Introduction

We study the task of mimicking a person's voice based on samples that are captured in-the-wild. As far as we know, no other solution exists for this highly applicable learning problem. While the current systems are mostly based on carefully collected or curated audio samples, our method is able to employ the audio of public speeches (from youtube), despite a large amount of background noise and clapping and even with an inaccurate automatic transcript. Moreover, almost all in-the-wild videos contain multiple other speakers that become challenging voice sample outliers and, in some cases, the videos are shot with home equipment and are of reduced quality.

Our method, called VoiceLoop, is inspired by a working-memory model known as the phonological loop (Baddeley, 1986). The loop holds verbal information for short periods of time. It comprises both a phonological store, where information is constantly being replaced, and a rehearsal process, which maintains longer-term representations in the phonological store.

In our method, we construct a phonological store by employing a shifting buffer that is best seen as a matrix $S \in \mathbb{R}^{d \times k}$ with columns $S[1] \ldots S[k]$. At every time point, all columns shift to the right ($S[i+1] = S[i]$ for $1 \leq i < k$), column $k$ is discarded, and a new representation vector $u$ is placed in the first position ($S[1] = u$). $u$ is a function of four parameters, among which are the latest "spoken" output and the buffer $S$ itself. The buffer is, therefore, constantly refreshed with new information, similar to the phonological store, and the mechanism that creates the representations reuses the existing information in the buffer, thus creating long term dependencies.

The two other input parameters of the network that computes the new representation $u$ are the identity of the speaker and the current attention-mediated context. The identity is captured by a learned embedding and is stored in a lookup table (for the individuals in the training set) or fitted (for new individuals). The usage of this embedding for the phonological store means that it influences the dynamic behavior of the store, the attention mechanism and the output process. Since the last process requires heavy personalization, it also receives the identity embedding directly.

---

[1] PyTorch code and sample audio files are available here: `https://github.com/facebookresearch/loop`

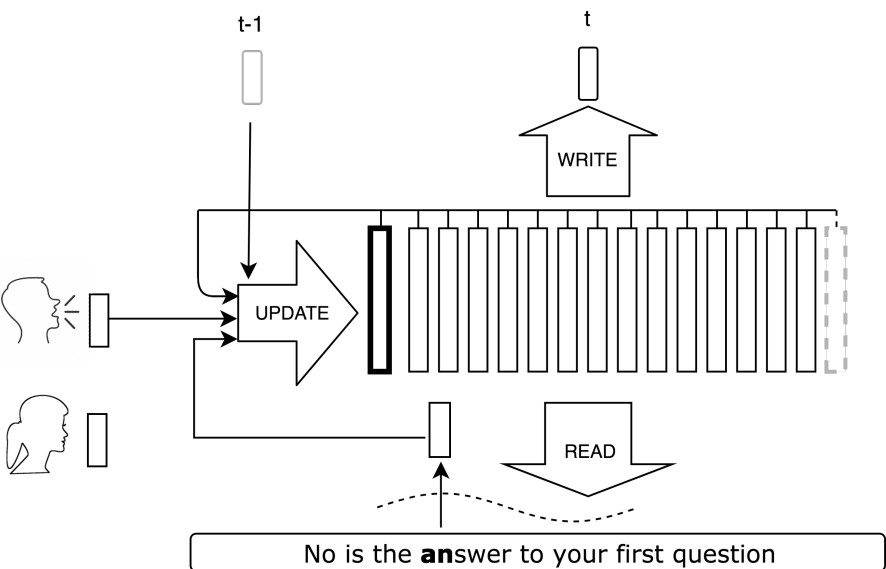

Figure 1: An overview of the VoiceLoop architecture. The reader combines the encoding of the sentence's phonemes using the attention weights to create the current context. A new representation is created by a shallow network that receives the context, the speaker ID, the previous output, and the buffer. The new representation is inserted into the buffer and the earliest vector in the buffer is discarded. The output is obtained by another shallow network that receives the buffer and the speaker as inputs. Once trained, fitting a new voice is done by freezing the network, except for the speaker embedding.

The input sentences in our system are represented as a list of phonemes. Each phoneme out of the 42 in the dictionary being employed, is encoded as a short vector. The encoding of an input sentence is the list of vectors which corresponds to its list of phonemes. The context, either through a Recurrent Neural Network (RNN) or triphones, is not used.

At each time point, the encodings of the phonemes are weighted and then summed, using a vector of attention weights, to form the current context vector. As attention mechanism, we employ the Graves attention model (Graves, 2013), which ensures a monotonic increase in the position along the sequence of input phonemes.

A few properties of our methods stand out in the landscape of neural text to speech work: (i) Instead of conventional RNNs, we propose to employ a memory buffer. (ii) The same memory is shared between all processes and is repeatedly used to make all inferences. (iii) We employ shallow fully-connected networks for all computations. (iv) The input encoding part of the "reader" mechanism is extremely simple.

We hypothesize that these properties make our architecture more robust than existing methods and allow us to mimic speakers based on noisy and limited training data. Moreover, since the output is more directly linked to the inputs, we are able to fit new speakers using relatively short audio sequences coupled with automatically generated text.

Finally, the output of our system is deterministic, given its input. However, multiple intonations are readily generated by employing priming, which involves initializing the buffer $S$ prior to the synthesis process.

Experimentally, we evaluate our method in two ways. For TTS quality, we follow the standard Mean Opinion Score (MOS) experiment done by Arik et al. (2017a). For speaker identification, we train a multi-class network which achieves near-perfect performance on a real validation set, and test it against generated ones.

## 2 PREVIOUS WORK

Text to speech (TTS) methods can be mostly classified into four families: rule-based, concatenative, statistical-parametric (mostly HMM based), and neural. HMM-based methods (Zen et al., 2009) require careful collection of the samples, or as recently attempted by Baljekar & Black, filtering of noisy samples for in-the-wild application. Concatenative methods are somewhat less restrictive but still require tens of minutes of clean and well transcribed samples from the target voice. Emerging neural methods may hold the (currently unrealized) promise of allowing the imitation of new speakers, based on limited and unconstrained samples captured in the wild.

Very recent neural TTS systems include the Deep Voice systems DV1 (Arik et al., 2017b) & DV2 (Arik et al., 2017a), WaveNet (Oord et al., 2016), Char2Wav (Sotelo et al., 2017), and Tacotron (Wang et al., 2017). The **DV2** system is a well-engineered system, which includes specialized subsystems for segmenting phonemes, predicting phoneme duration, and predicting the fundamental frequency. Each subsystem includes stacked bidirectional recurrent networks, multilayer fully connected networks and many residual connections. This stands in stark contrast to our system, which employs a single shared memory, one output process, and shallow fully connected networks.

DV2 is the only other current method that models multiple speakers in a single network. However, in contrast to our results, there are three critical differences: (a) There are no in-the-wild experiments; (b) no fitting to a new speaker that did not appear in the training set is shown possible; and (c) the authors employ a large private set and delegate the attention problem to sub-systems, including strong ground-truth alignment between phonemes, waveforms and linguistic features. The linguistic features, which comprise of phone duration, syllable stress, number of syllables in a word and position of the current syllable in a phrase, are also used during inference for generating the samples (used in the subjective Mean Opinion Score tasks as well). In contrast, our method learns "where to read" from the input. Note that (a) and (b) are crucial capabilities in making TTS accessible to a wide range of applications, in particular when casually and efficiently modeling non-professional speakers. The need for professionally collected datasets and the lack of post-training fitting could be inherent to the DV2 architecture, since it has a large number of speaker-dependent modules, whereas we fit a new speaker in a single place.

The **Tacotron** system employs a multi-stage encoder-decoder architecture with multiple RNNs and a block called CBHG (Lee et al., 2016) components, with each CBHG containing multiple convolutional layers, a highway network (Srivastava et al., 2015), and a bidirectional GRU (Cho et al., 2014). The output is a synthesized spectrogram, from which the audio is reconstructed by the Griffin-Lim (Griffin & Lim, 1984) method. Trained on a large private training set recorded by a professional single speaker, the Tacotron system is able to read raw text (characters and not phonemes). While Tacotron was not trained for multiple speakers, Arik et al. (2017a) have done so and report a high level of sensitivity to the choice of parameters and a need to incorporate the input embedding in many network sites. The **Char2Wav** architecture employs RNNs for both the reader and the generator. As an attention mechanism, the Graves positional attention mechanism (Graves, 2013) is used. The same attention mechanism is used in our work. However, in our case, the parameters of the attention model are based on the shared memory store (the buffer). Similarly to our method, the network was also trained to predict vocoder features. In addition, for added quality, the vocoder was replaced by a SampleRNN network (Mehri et al., 2016). In contrast to the above mentioned systems, which employ RNNs, the **WaveNet** architecture is based on stacks of dilated convolutions, which are termed "causal" for not looking into the future. The output audio is generated sample by sample, which, at typical sampling rates of thousands of hertz, is too slow for current TTS applications. Wavenet has shown single-speaker TTS capabilities, but not multi-speaker.

**Waveforms Synthesis** There is currently no TTS method which can synthesize waveforms from scratch. WaveNet, DV1, DV2, Char2wav and Tacotron were all conditioned on top of lower level generators. Wavenet was conditioned on F0 vocoder features, as well as linguistic features extracted from separately trained RNN-based text representations. SampleRNNs were employed on top of vocoders. Tacotron synthesized spectograms from mel-spectograms, approximating waveforms using Griffin-Lim. As observed by DV2, small errors in the spectrogram generation result in unnatural (metallic) noise in the reconstruction. Further audio processing can be used to alleviate them, but to a limited extent. Better results were achieved (Arik et al., 2017a) by replacing Griffin-Lim with a Wavenet-like net conditioned on the generated spectogram and speaker.

Table 1: The components of the VoiceLoop model

| | Symbol | Description | Computed as: |
|---|---|---|---|
| **Variables** | $S_t \in \mathbb{R}^{d \times k}$ | buffer at time $t$ | $S_t[1] = u_t; S_t[i+1] = S_{t-1}[i]$ |
| | $u_t \in R^d$ | new representation for the buffer | $N_u([S_{t-1}, [c_t + tanh(F_u z), o_{t-1}]])$ |
| | $E \in \mathbb{R}^{d_p \times l}$ | embedding of the input sequence | $E[i] = LUT_p[s_i]$ |
| | $z \in \mathbb{R}^{d_s}$ | embedding of the current speaker | $LUT_s[id]$ or Sec. 3.2 |
| | $\kappa_t, \beta_t, \gamma_t \in \mathbb{R}^c$ | attention model parameters | $N_a(S_{t-1})$ |
| | $\mu_t, \sigma_t^2, \gamma_t' \in \mathbb{R}^c$ | attention GMM parameters | $\mu_t = \mu_{t-1} + e^{\kappa_t}, \sigma_t^2 = e^{\beta_t}, \gamma_t' = \text{sm}(\gamma_t)$ |
| | $\alpha_t \in \mathbb{R}^l$ | attention vector at time $t$ | See Eq. 3, 4 |
| | $c_t \in \mathbb{R}^{d_p}$ | context vector at time $t$ | $c_t = E\alpha_t$ |
| | $o_t \in \mathbb{R}^{d_o}$ | output vector at time $t$ | $N_o(S_t) + F_o z$ |
| **Networks** | $N_u : kd + d_p + d_o \to d$ | buffer update network | |
| | $N_a : kd \to 3c$ | attention network | |
| | $N_o : kd \to d_o$ | output network | |
| | $LUT_p \in \mathbb{R}^{d_p \times 42}$ | embedding of each phoneme | |
| | $LUT_s \in \mathbb{R}^{d_s \times N}$ | embedding of the speakers | |
| | $F_u : d_s \to d_p$ | projection of the speaker for update | |
| | $F_o : d_s \to d_o$ | projection of the speaker for output | |
| **Parameters** | $d$ | dimensionality of the buffer | $d_p + d_o$ |
| | $k$ | capacity of the buffer | 20 |
| | $d_p$ | dim. of the input embedding LUT | 256 |
| | $d_o$ | dim. of the vocoder feature vector | 63 |
| | $d_s$ | dim. of the speaker embedding | $d_p$ |
| | $c$ | # GMM component (attention model) | 10 |
| | $s_1 \dots s_l, 1 \le s_i \le 42$ | input sequence | |
| | $l$ | length of the input sequence | |
| | $N$ | number of speakers in the training set | |

Our system was designed with simplicity in mind in order to promote robustness and reproducibility. We focus on modeling the underlying generation process and do not integrate or condition explicitly for waveforms synthesis. Instead, we employ the WORLD (Morise et al., 2016) vocoder (D4C edition) for feature extraction and waveform synthesis. While this bounds the achievable quality, we also experimented with adding WaveNet and SampleRNN. However, the added performance did not seem to justify the extra effort, especially for in-the-wild voice training data, where we observed no improvement.

**Differentiable Memory** The differentiable buffer architecture, in which a new representation is added at every step, and the last vector added is discarded in a FIFO manner, is novel as far as we know. There are multiple other network models in the literature that are augmented by an external memory structure, e.g., (Joulin & Mikolov, 2015; Sukhbaatar et al., 2015; Graves et al., 2014). However, to our knowledge, our work is one of very few applications of such memory networks outside in practice.

Perhaps the closest model to our work is Stack RNN by Joulin & Mikolov (2015), in which the network is augmented with an infinite stack to which a state vector can be added (PUSH) or removed (POP) at every time step. Unlike our model, only the top of the stack is read each time.

## 3 THE ARCHITECTURE

The architecture of the VoiceLoop model is depicted in Fig. 1 and the components of the architecture are listed in Tab. 1. The forward pass of the network has four steps, which are run sequentially. Following a context-free encoding of the input sequence and an encoding of the speaker, the buffer at time t, $S_t \in \mathbb{R}^{d \times k}$, plays a major role in all of the remaining steps and links between the other components of each step. It also carries the error signal from the output to the earlier steps.

**Step I: Encoding the speaker and the input sentence**    Every speaker is represented by a vector $z$. During training, the vectors of the training speakers are stored in a lookup table $LUT_s$ which maps a running id number to a representation of dimensionality $d_s$. For new speakers, which are being fitted after the network was trained, the vector $z$ is computed by the straightforward optimization process described in Sec. 3.2.

The input sentence is converted to a sequence of phonemes $s_1, s_2, \ldots, s_l$ by employing the CMU pronouncing dictionary (Weide, 1998). The number of phonemes in this dictionary is 40, to which two items are added to indicate pauses of different lengths. Each $s_i$ is then mapped separately to an encoding that is based on a trained lookup table $LUT_p$. This results in an encoding matrix $E$ of size $d_p \times l$, where $d_p$ is the size of the encoding, and $l$ is the sequence length.

**Step II: Computing the context**    Similar to (Sotelo et al., 2017; Chorowski et al., 2015), we employ the Graves Gaussian Mixture Model (GMM)-based monotonic attention mechanism. At each output time point $t = 1, 2, \ldots$, the attention network $N_a$ receives the buffer from the previous time step $S_{t-1}$ as input and outputs the GMM priors $\gamma_t$, shifts $\kappa_t$, and log-variances $\beta_t$. For a GMM with $c$ components, each of these is a vector in $\mathbb{R}^c$. $N_a$ has one hidden layer, of dimensionality $\frac{dk}{10}$ and a ReLU activation function for the hidden layer.

The attention is then computed as follows:

$$\gamma_t'[i] = \frac{exp(\gamma_t[i])}{\sum_j exp(\gamma_t[j])}, i = 1, 2, \ldots, c \tag{1}$$

i.e., the softmax function is applied to the priors. The means of the GMMs are increased:

$$\mu_t = \mu_{t-1} + exp(\kappa_t), \tag{2}$$

and the variances are computed as $\sigma_t^2 = exp(\beta_t)$. For each GMM component $1 \le i \le c$ and each point along the input sequence $1 \le j \le l$, we then compute:

$$\phi[i, j] = \frac{\gamma_t'[i]}{\sqrt{2\pi\sigma_t^2[i]}} exp(-\frac{(j - \mu_t[i])^2}{2\sigma_t^2[i]}) \tag{3}$$

The attention weights $\alpha_t$ are computed for each location in the sequence by summing along all $c$ components:

$$\alpha_t[j] = \sum_{i=1}^{c} \phi[i, j] \tag{4}$$

The context vector $c_t$ is then computed as weighted sum of the columns of the input sequence embedding matrix $E$ as $c_t = E\alpha_t$. The loss function of the entire model depends on the attention vector through this context vector. The GMM is differentiable with respect to mean, std and weight, and these are updated, during training, through backpropagation.

**Step III: Updating the buffer**    At each time step, a new representation vector $u$ of dimensionality $d$ is added to the buffer at the first location $S_t[1]$, the last column of the buffer from the previous time step $S_{t-1}[k]$ is discarded, and the rest are copied $S_t[i + 1] = S_{t-1}[i]$ for $i = 1, \ldots, k - 1$.

In our implementation, the number of features in the buffer $d$ is the sum of the dimensionality of the embedding of the phonemes $d_p$ and the output's dimensionality $d_o$. This choice was made so that a direct comparison to a buffer that does not employ an update network can be performed. In this case, $u$ is simply the concatenation of the current context vector $c_t$ and the output from the previous time step $o_{t-1}$. It soon became very clear that this loop-less buffer update leads to poor results, emphasizing the role of using information of the buffer $S$ itself in the update process.

The vector $u$ is, therefore, computed using a shallow fully connected network $N_u$, with one hidden layer of a size that is the tenth of the input dimensionality and a ReLU activation function.

The network receives as input the buffer $S_{t-1}$, the context vector $c_t$, and the previous output $o_{t-1}$. The new vector $u$ is also made speaker dependent by adding a projection of the speaker embedding $z$ to the context vector. This projection is followed by a hyperbolic tangent activation function, in order to maintain scale. Therefore,

$$C_t = [c_t + tanh(F_u z), o_{t-1}] \tag{5}$$

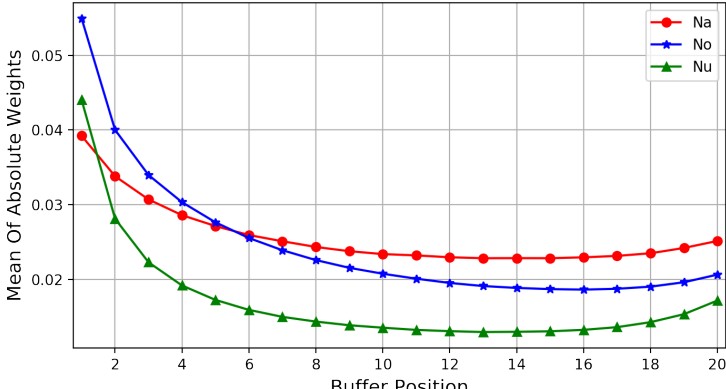

Figure 2: Memory Location Significance. For each of the three networks $N_u$, $N_a$ and $N_o$, we average the absolute values of the weights to the hidden layer across all hidden neurons and across the $d$ rows of the buffer. The result is a measure of the relative importance of each column of the buffer. Best viewed in color.

$$u = N_u([S_{t-1}, C_t]), \tag{6}$$

where $[a, b]$ is the concatenation of the two column vectors $a$ and $b$ to one column vector, or the concatenation of two matrices $a$ and $b$ side by side.

Another way in which we allow the speaker to influence the generated output is by initializing the buffer based on the speaker's embedding. Specifically, in our implementation, the speaker embedding size $d_s$ is the same as the phoneme embedding size $d_p$ and we set the top part of the buffer $S_0$ to be $z$ repeated $k$ times. The lower part of size $d_o \times k$ is set to zero.

**Step IV: Generating the output**    The output is generated using a network $N_o$ that is of the same architecture as $N_a$ and $N_u$ and a projection of the user by a learned matrix $F_o$:

$$o_t = N_o(S_t + F_o z) \tag{7}$$

**Memory Location Significance**    In order to better understand the behavior of the buffer, we consider the relative role of each buffer location $1, 2, \ldots, k$ on the activations of $N_u$, $N_a$, and $N_o$. Specifically, we average the absolute values of the weights from the input (buffer elements) to the hidden layer. The averaging is performed across all $d$ features and $\frac{dk}{10}$ hidden units, and provides one value per each location. As can be seen in Fig 2, the weights of the latest elements are more prominent, especially, as expected, for the output network $N_o$. However, even the rightmost column has a relative contribution that is at least one third of the leftmost column. This supports the utility of our buffer architecture, in which all memory locations are equal inputs to the downstream fully connected networks.

## 3.1    Training

In our current implementation, the output is a vector of vocoder features of dimensionality $d_o = 63$. Similar to (Sotelo et al., 2017), these features were computed using the Merlin toolkit (Wu et al., 2016). During training, the output at each time frame $t$ is compared to the vocoder features of the ground truth data $Y_t$ using the MSE loss: $\frac{1}{d_o}\|Y_t - o_t\|^2$. This loss requires an exact temporal alignment of the input and the output sequence. However, human speech is not deterministic and one cannot expect a deterministic method to predict the ground truth. For example, even the same speaker cannot replicate her voice to completely remove the MSE loss since there is variability when repeating the same sentence. Teacher forcing solves this since it eliminates most of the drift and enforces a specific way of uttering the sentence.

In conventional teacher forcing, during training, the input to the network $N_u$ is $Y_{t-1}$ and not $o_{t-1}$. This holds the danger of teaching the network to predict only one time frame ahead, which would

create a drift in the output when run on test data. We, therefore, employ a variant of the teacher-forcing technique, which uses the following input to $N_u$ as the previous output

$$\frac{o_{t-1} + Y_{t-1}}{2} + \eta, \tag{8}$$

where $\eta$ is a random noise vector. When training starts, the predicted output $o_{t-1}$ is by itself a source of noise. As training progresses, it becomes more similar to $Y_{t-1}$. However, the systematic difference between the two allows the network to better fit the situation that occurs at test time.

During training, a forward pass on all of the output sequences is performed (without truncation), followed by a backward pass.

**Efficiency**  The full model contains 9.3 million parameters and runs near real-time on an Intel Xeon E5 single-core CPU and 5 times faster when on M40 NVIDIA GPU, including vocoder CPU decoding. This was benchmarked with our publicly available python PyTorch implementation. Therefore, even without special optimizations, engineering VoiceLoop to run on a mobile client is possible, similar to existing non-neural TTS client solutions (e.g. Android's text-to-speech APK).

## 3.2 FITTING A NEW PERSON

Different people exhibit different patterns and present various mannerisms in their speech. Therefore, learning to fit these factors from a limited amount of speech is a challenging task. The goal of speaker mimicking TTS is to be able to mimic a new person based on a relatively short voice sample. Ideally, the new voice would be captured by the parameters of the speaker embedding $z$, without the need to retrain the network. Naturally, enough variability in the population of the training speakers is needed in order to support this. To fit a new speaker, we are given voice samples and transcribed text. We then employ the training procedure, where the weights of all networks and projections ($N_a, N_u, N_o, LUT_p, F_u, F_o$) are kept fixed and only vector $z$ is learned (using SGD) to form the embedding of the new speaker.

The same training procedure as detailed in Sec. 3.1 is employed for fitting a new person, including the application of teacher-forcing. We find that the fitting process is very stable with regards to voice characteristics such as pitch. We also noticed that the accent in the new sample needs to be relatively close to the accents presented in the training samples. See Sec. 3.2 for fitting experiments.

## 3.3 GENERATING VARIABILITY

As mentioned, natural speech is not deterministic and each time a sentence is said, it is said in a different way. For simplicity, our method does not employ a random component, such as a variational autoencoder. However, we can generate different outputs by employing priming (Graves, 2013). In this technique, the initial buffer $S_0$ is initialized based on an initial process in which another word or sentence is run through the system. One can expect that a sentence from the training set that is said in excitement, would paint the buffer differently than one that is flatter. Experimenting with this technique, demonstrates that we are indeed able to achieve the desired level of variability. However, the direct link between the nature of the priming sequence and the generated output is only anecdotal at this point.

## 4 EXPERIMENTS

We make use of multiple datasets. First, for comparing with existing single speaker techniques, we employ single speaker literature datasets. Second, we employ various subsets of the VCTK dataset (Veaux et al., 2017) for various multi-speaker training and/or fitting experiments. Third, we create a dataset that is composed from four to five public speeches of four public figures. The data was downloaded from youtube, where these speeches are publicly available and were automatically transcribed. Samples generated by our method are available on the project's website `https://github.com/facebookresearch/loop`.

The MOS measure for the proposed method was computed using the crowdMOS toolkit by P. Ribeiro et al. (2011) and Amazon Mechanical Turk. All samples were presented at 16kHz and the raters were told that they are presented with the results of the different algorithms. At least 20 raters participated

in each such experiment, with 95% confidence intervals. We restricted all experiments to North American raters.

## 4.1 SINGLE SPEAKER EXPERIMENTS

The single speaker experiments took place on the LJ (Ito, 2017a), the Nancy corpus from the 2011 Blizzard Challenge (King & Karaiskos, 2011), and the English audiobook data for the 2013 Blizzard Challenge (King & Karaiskos, 2013). Our method was compared to the ground truth as well as to Char2Wav and to Tacotron. The Char2Wav system was trained by us using the authors' implementation available at `https://github.com/sotelo/parrot`. The training of the Char2Wav model, in each experiment, was optimized by measuring the loss on the validation set, over the following hyperparameters: initial learning rate of $[1e-2, 1e-3, 1e-4]$, source noise standard deviation ($[1, 2, 4]$), batch-size ($[16, 32, 64]$) and the length of each training sample ($[10e2, 10e4]$).

The Tacotron models were pretrained models available from the best public implementation we could find, which is by Ito (2017b). This re-implementation has models only for the LJ and the Nancy datasets. Note that Tacotron has raised a lot of attention and considerable effort was put by the community to replicate the paper's results. However, there would very likely be a different choice of hyper-parameters between such re-implementations and the one of the authors.

The MOS scores are shown in Tab. 2. These were computed using the "same_sentence" option of crowdMOS, following DV2 (personal communication). As can be seen, our single speaker results are better than those of the other two algorithms across datasets, but still somewhat lower than the ground truth results.

It is interesting to note that on Blizzard 2011, our results are better than Tacotron (reimplementation) but not significantly better than Char2Wav, while on Blizzard 2013 it is significantly better than both. This can be attributed to the clean nature of Blizzard 2011, for which Char2Wav is robust enough, and demonstrates our method's robustness to noise.

Tab. 3 presents Mel Cepstral Distortion (MCD) scores. This is an automatic, albeit limited, method of testing compatibility between two audio sequences. Since the sequences are not aligned, we employ MCD DTW, which uses dynamic time warping (DTW) to align the sequences. As can be seen, in this metric too, our method outperforms the baseline methods. The single except is Tacotron's lower distortion on the LJ dataset. However, as shown in Tab. 2, Tacotron is not competitive on this dataset.

Table 2: Single Speaker MOS Scores (Mean $\pm$ SD)

| Method | LJ | Blizzard 2011 | Blizzard 2013 |
|---|---|---|---|
| Tacotron (re-impl) | $2.06 \pm 1.02$ | $2.15 \pm 1.10$ | N/A |
| Char2wav | $3.42 \pm 1.14$ | $3.33 \pm 1.06$ | $2.03 \pm 1.16$ |
| VoiceLoop | $3.69 \pm 1.04$ | $3.38 \pm 1.00$ | $3.40 \pm 1.03$ |
| Ground truth | $4.60 \pm 0.71$ | $4.56 \pm 0.67$ | $4.80 \pm 0.50$ |

Table 3: Single Speaker MCD Scores (Mean $\pm$ SD; lower is better)

| Method | LJ | Blizzard 2011 | Blizzard 2013 |
|---|---|---|---|
| Tacotron (re-impl) | $12.82 \pm 1.41$ | $14.60 \pm 7.02$ | N/A |
| Char2wav | $19.41 \pm 5.15$ | $13.97 \pm 4.93$ | $18.72 \pm 6.41$ |
| VoiceLoop | $14.42 \pm 1.39$ | $8.86 \pm 1.22$ | $8.67 \pm 1.26$ |

## 4.2 MULTI-SPEAKER EXPERIMENTS

Multi-speaker experiments were performed on the VCTK dataset (Veaux et al., 2017). The 109 speakers were divided into four different nested subsets: 22 North American speakers, both male and females; and 65, 85 and 101 random selection of speakers, where the remaining eight speakers were left out for validation. Each subset was shuffled into train and test sets. Different models were trained to each of the subsets. Qualitatively, the models provide distinguished voices, and as can be seen in Fig. 3, the generated voice samples display a different dynamic behavior for different speakers.

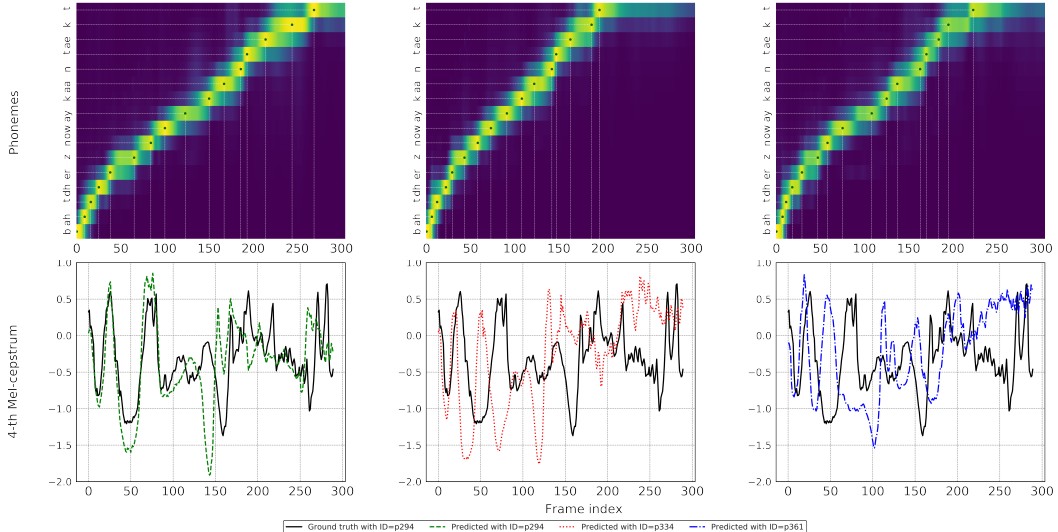

Figure 3: Top: The attention probabilities obtained when mimicking three different North American speakers from VCTK using *the same* sentence: "but there is no eye contact". The x-axis is the time along the generated audio. The y-axis depicts the sequence of phonemes. Dots indicate the maximal response along time for each phoneme, illustrating *learned* phoneme duration differences between identities (not given during training). Bottom: The 4-th Mel-cepstrum for the three generated sentences (dashed) as well as the ground-truth (solid) of the leftmost speaker. Best viewed in zoom.

Table 4: Multi-speaker MOS scores (Mean ± SE)

| Method | VCTK22 | VCTK65 | VCTK85 | VCTK101 |
|---|---|---|---|---|
| Char2wav | $2.84 \pm 1.20$ | $2.85 \pm 1.19$ | $2.76 \pm 1.19$ | $2.66 \pm 1.16$ |
| VoiceLoop | $3.57 \pm 1.08$ | $3.40 \pm 1.00$ | $3.13 \pm 1.17$ | $3.33 \pm 1.10$ |
| GT | $4.61 \pm 0.75$ | $4.59 \pm 0.72$ | $4.64 \pm 0.64$ | $4.63 \pm 0.66$ |

In our experiments, we employ the author's implementation of Char2Wav mentioned above as baseline. Note that while the Char2Wav paper did not present multi-speaker results, the open implementation is more general and includes this option.

Sentences from the test set of VCTK are employed for testing. Following DV2 (private communication), the MOS results were computed using the "diff_sentences" option of the crowdMOS toolkit, and are depicted in Tab. 4. As can be seen, our multi-speaker method shows a considerable advantage over the Char2Wav system across all VCTK subsets, but is not as good as the ground truth. These results are consistent with the MCD scores as reported in Tab. 5.

**Speaker Identification** The capability of the system to generate distinguished voices that match the original voices was tested, as was done in DV2, using a speaker classifier. We train a multi-class convolutional network on the ground-truth training set of multiple speakers, and test on the generated ones. The network gets as input an arbitrary size of vocoder samples, performs five convolutional layers of 3x3 filters over 32 batch-normalized channels, followed by max-pooling, average pooling over time, two fully-connected layers, and ending with a softmax of the number of classes tested. All intermediate layers were linearly rectified.

The identification results are shown in Tab. 6. The VoiceLoop results are more accurate than the results on the VCTK test split, despite using the same text. This might indicate that the voices generated are more similar to the training voices than the natural variability that is present in the dataset. The Char2Wav results are considerably lower.

Table 5: Multi-speaker MCD scores (Mean $\pm$ SE; lower is better)

| Method | VCTK22 | VCTK65 | VCTK85 | VCTK101 |
|---|---|---|---|---|
| Char2wav | $15.71 \pm 1.82$ | $15.1 \pm 1.45$ | $15.23 \pm 1.49$ | $15.06 \pm 1.32$ |
| VoiceLoop | $13.74 \pm 0.98$ | $14.1 \pm 0.94$ | $14.16 \pm 0.87$ | $14.22 \pm 0.88$ |

Table 6: Multi-Speaker Identification Top-1 Accuracy (%)

| Method | VCTK85 | VCTK101 |
|---|---|---|
| VCTK test split | 98.25 | 97.16 |
| Char2Wav on test split sentences | 75.70 | 81.63 |
| VoiceLoop on test split sentences | 100 | 99.76 |

### 4.3 New speaker fitting experiments

Our system is the only published system that is capable of post-training fitting of new speakers. In order to experiment with this capability, we employ the VoiceLoop model trained on VCTK85 and experiment on the remaining 16 speakers one by one, where only the speaker embedding $z$ gets updated. While TTS systems typically require several hours of data to model a single speaker (Zen et al., 2009), our fitting set contains only 23.65 minutes per speaker on average.

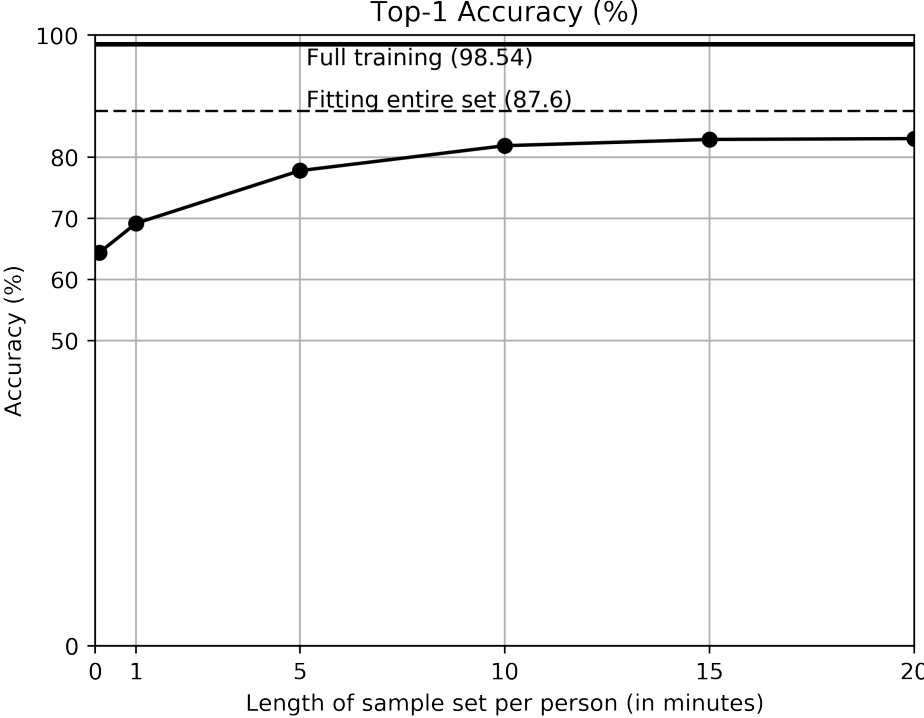

Figure 4: Fitting new speaker embeddings to an existing VoiceLoop model. The graph plots top-1 identification accuracy with respect to a sample set length (in minutes) per speaker. Scores were averaged over 5 splits each. The "Full training" horizontal line is the top-1 accuracy for the corresponding speakers, when trained together with the model from scratch. The leftmost datapoint is for two sentences (about 10sec) per speaker.

As described in 3.2, we randomly initialize a new embedding for every new speaker and update only its weights during back-propagation on the fitting data. The newly fitted speakers achieve **3.08 ± 0.95** MOS, suggesting that the generation mechanism has not deteriorated below a "fair" level by the new entries.

Similar to the multi-speaker case, we train classifiers for the corresponding identities on ground-truth data, but test on the fitted ones, achieving **87.6%** top-1 identification accuracy. Despite lower rates than those in Tab. 6, generations of fitted identities are still reasonably discriminative. We conjecture that training VoiceLoop on a larger set of speakers (e.g. LibriSpeech Panayotov et al. (2015)) will be able to represent unseen identities better.

**Fitting Data Size**    The performance of fitting a new identity clearly relies on the length of the sample that is available for that speaker. In order to understand the influence of the sample size, we repeated the above fitting process for the 16 speakers, but capped the available fitting data per speaker. Specifically, we experimented with a maximal amount of training data of 1, 5, 10, 15 and 20 minutes of voice for each speaker. Instead of cutting the last sentence in the middle, it was removed in case that the threshold was crossed. We repeated this fitting process 5 times, each time fitting a different set of samples at a particular limit.

In Fig. 4 we report identification accuracies for each limit. Surprisingly, even with two sentences per speaker, totaling about 10 seconds in average, we can fit a new speaker into VoiceLoop such that the speaker is identifiable at **64.4%** top-1 identification rate.

### 4.4    IN THE WILD EXPERIMENTS

To demonstrate the flexibility of our method, we downloaded several publicly available videos from youtube. We picked four different known speakers (see samples page), and for each we retrieved the top four to five results, provided that they are longer than 20 minutes. We extracted the audio and its associated (youtube's) automatically transcribed text. The total amount of data is 6.2 hours, which we then segmented into 8000 segments. Each segment length is around three seconds, similar to the datasets used in the experiments above. Both the data and its corresponding text are noisy: some of the samples include panel discussions and others with questions from various reporters. Sometimes, microphone echo was observed, or relatively low quality audio originated from mobile video conference sessions. We then trained on this data a VoiceLoop model from scratch, using exactly the same training procedure used by the other experiments. This achieved MOS is **2.97 ± 1.03**, and top-1 accuracy of **95.81%**.

We also demonstrate priming (Sec. 3.3) on this dataset. Even for the same speaker, multiple intonations can be generated by initializing $S_0$ in different ways. This capability is depicted in Fig. 5 and in the samples page.

## 5    DISCUSSION

Employing web-based in-the-wild training data means that the network is trained on mixed data that contains both speech and other sources. For example, our samples contain a considerable amount of clapping and laughs. Moreover, public speeches contain a larger than usual amount of dramatic prosody and methodological pauses (the same is also true with audiobooks). As our experiments show, our method is mostly robust to these, since it is able to model the voices despite of these difficulties and without replicating the background noises in the synthesized output. The baseline model of Char2Wav was not able to properly model the voices of the youtube dataset and presented clapping sounds in its output.

The architectural simplicity of our system is likely to be the reason for its robustness. Another advantage that stems directly from it, is its computational efficiency. Based on a few shallow networks and on an iterative process that does not consider future samples, our method can generate voice on mobile devices in speeds far exceeding real-time. For comparison, deep voice (Arik et al., 2017b) is posed as a real time neural TTS system, and it achieves a rate of up to 2.7 times real-time on a Intel Xeon E5-2660 v3 Haswell CPU, running 6 concurrent threads (GPU does not provide speedup for the inference of the deep voice system).

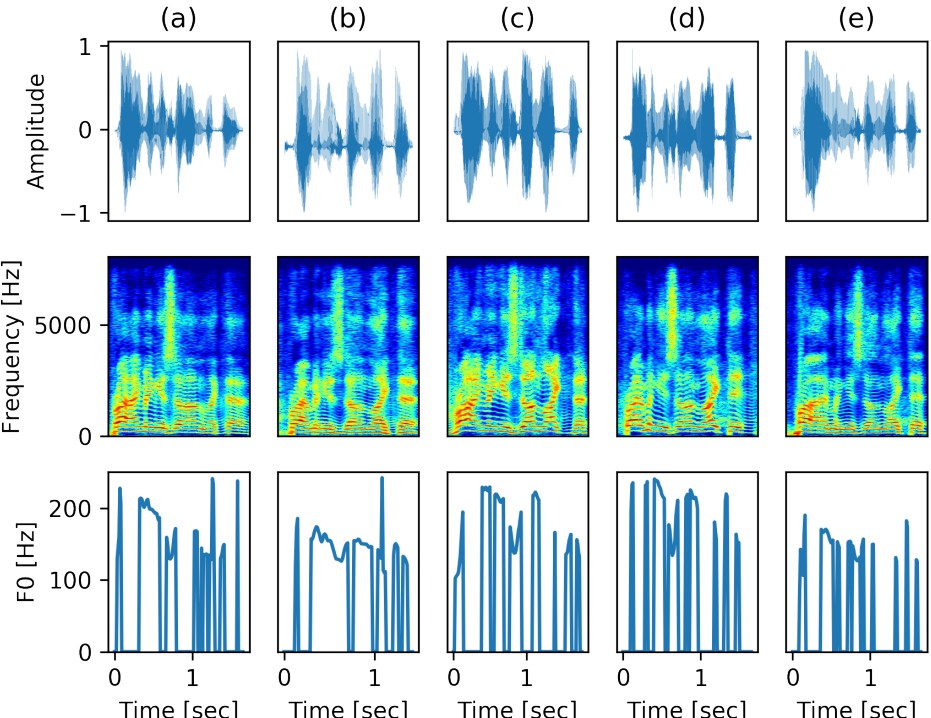

Figure 5: Same input, different intonations. A single in the wild speaker saying the sentence "priming is done like that ", where each time $S_0$ is initialized differently. (a) Without priming. (b) Priming with the word "I". (c) Priming with the word "had". (d) Priming with the word "must". (e) Priming with the word "bye". The figure shows the raw waveform, spectrogram, and F0 estimation (include voicedness) in the first, second and third rows respectively. From the spectrogram plots we can observe different duration for some phonemes. The F0 estimation of (c) and (d) shows that the speaker talks in higher tone while in (b) and (e) we can observe lower tone of the speaker. This demonstrates how priming changes the intonations of the model outputs.

The link we form to the model of Baddeley (1986) is by way of analogy and, to be clear, does not imply that we implement this model as is. Specifically, by phonological features, we mean a joint (mixed) representation, in memory, of sound based information and language based information, which is a unique characteristic of our model in comparison to previous work. The short term memory in Baddeley's model is analog to our buffer and the analog to the rehearsal mechanism is the recursive way in which our buffer is updated. Namely, the new element in the buffer ($u$) is calculated based on the entire buffer. As noted in Sec. 3, without this dependency on the buffer, our model becomes completely ineffective.

While we employ the loop-updated buffer for the task of speech synthesis, the model is quite general. For example, we have employed the buffer for machine translation from English to French using a dot product based attention model (Bahdanau et al., 2014). The discrete nature of the output means that an output embedding had to be added, but the overall structure remained the same. The performance seemed at least similar to the baseline RNN attention model. However, no attempt has yet been made to achieve state of the art results on existing benchmarks. Surprisingly, relatively large buffer sizes (9) seem to produce better results, despite the input and the output being relatively short. Staying in the realm of voice, the buffer model can be readily used to form a transformation in the other direction (from speech to text), and applied to audio denoising.

# 6 CONCLUSION

We present a new memory architecture that serves as an effective working memory module. Building on this, we are able to present a neural TTS solution of an architecture that is less complex than those found in the recent literature. It also does not require any alignment between phonemes and acoustics or linguistic features as inputs. Using the new architecture, we are able to present, for the first time as far as we know, multi-speaker TTS that is based on unconstrained samples collected from public speeches. Our work also presents a unique ability to fit new speakers (post-training), which is demonstrated even for very limited sample size.

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
