# OpenReview forum: "VoiceLoop: Voice Fitting and Synthesis via a Phonological Loop"
_ICLR.cc/2018/Conference — Accept (Poster)_

### Official Review · AnonReviewer1 · 2017-11-22
**a good paper. accept.**

**Rating:** 8
**Confidence:** 4

**Review:**

This is an interesting paper investigating a novel neural TTS strategy that can generate speech signals by sampling voices in the wild.  The main idea here is to use a working memory with a shifting buffer.  I also listened to the samples posted on github and the quality of the generated voices seems to be OK considering that the voices are actually sampled in the wild. Compared to other state-of-the-art systems like wavenet, deep voice and tacotron, the proposed approach here is claimed to be simpler and relatively easy to deploy in practice.   Globally this is a good piece of work with solid performance. However, I have some (minor) concerns.

1.  Although the authors claim that there is no RNNs involved in the architectural design of the system,  it seems to me that  the working memory with a shifting buffer which takes the previous output as one of its inputs is a network with recurrence.

2. Since the working memory is the key in the architectural design of VoiceLoop, it would be helpful to show its behavior under various configurations and their impact to the performance. For instance,  how will  the length of the running buffer affect the final quality of the voice?

3. A new speaker's voice is generated by only providing the speaker's embedding vector to the system.  This will require a large number of speakers in the training data in the first place to get the system learn the spread of speaker embeddings in the latent (embedding) space.  What will happen if a new speaker's acoustic characteristics are obvious far away from the training speakers?  For instance, a girl voice vs. adult male training speakers.  In this case, the embedding of the girl's voice will show up in the sparse region of the embedding space of training speakers.  How does it affect the performance of the system?  It would be interesting to know.

---

> ### Author Response · Authors · 2017-12-12
> **a copy of our reply from 23 Nov**
>
> 1. The proposed memory model is indeed a network with recurrence and, therefore, it is a Recurrent Neural Network. We meant to highlight the differences from conventional and other existing RNN models. We will clarify this point.
>
> 2. In our experiments, buffer sizes between 15 and 30 seem to produce similar results. Less than 15 is detrimental. See also Fig. 2, which depicts the relative contribution of each column of the buffer and shows that all columns impact the computations done for the attention, the output, and the buffer update.
>
> 3. We have not tested the specific experiment that is described but have done a similar experiment in which we train on voices of north Americans and fit on other accents. The results confirm that when fitting out of distribution voices, the quality degrades. However, the fitting is still successful in capturing the pitch and other aspects of the voice.
>
> The authors.

---

### Official Review · AnonReviewer2 · 2017-11-27
**Interesting paper - not sure if ICLR is the right venue**

**Rating:** 5
**Confidence:** 4

**Review:**

This paper present the application of the memory buffer concept to speech synthesis, and additionally learns a "speaker vector" that makes the system adaptive and work reasonably well on "in-the-wild" speech data. This is a relevant problem, and a novel solution, but synthesis is a wicked problem to evaluate, so I am not sure if ICLR is the best venue for this paper. I see two competing goals:

- If the focus is on showing that the presented approach outperforms other approaches under given conditions, a different task would be better (for example recognition, or some sort of trajectory reconstruction)
- If the focus is on showing that the system outperforms other synthesis systems, then a speech oriented venue might be best (and it is unfortunate that optimized hyper-parameters for the other systems are not available for a fair comparsion)
- If fair comparisons with the other appraoches cannot be made, my sense is that the multi-speaker (post-training fitting) option is really the most interesting and novel contribution here, which could be discussed in mroe detail

Still, the approach is creative and interesting and deserves to be presented. I have a few questions/ suggestions:

Introduction

- The link to Baddeley's "phonological loop" concept seems weak at best. There is nothing phonological about the features that this model stores and retrieves, and no evidence that the model behaves in a way consistent with "phonologcial" (or articulatory) assumptions or models - maybe best to avoid distracting the reader with this concept and strengthen the speaker adaptation aspect?
- The memory model is not an RNN, but it is a recurrently called structure (as the name "phonological loop" also implies) - so I would also not highlight this point much
- Why would the four properties of the proposed method (mid of p. 2, end of introduction: memory buffer, shared memory, shallow fully connected networks, and simple reader mechanism) lead to better robustness and improve performance on noisy and limited training data? Maybe the proposed approach works better for any speech synthesis task? Why specifically for "in-the-wild" data? The results in Table 2 show that the proposed system outperforms other systems on Blizzard 2013, but not Blizzard 2011 - does this support the previous argument?
- Why not also evaluate MCD scores? This should be a quick and automatic way to diagnose what the system is doing? Or is this not meaningful with the noisy training data?

Previous work

- Please introduce abbreviations the first time they are used ("CBHG" for example)
- There is other work on using "in-the-wild" speech as well: Pallavi Baljekar and Alan W Black. Utterance Selection Techniques for TTS Systems using Found Speech, SSW 2016, Sunnyvale, USA Sept 2016

The architecture
- Please explain the "GMM" (Gaussian Mixture Model?) attention mechanism in a bit more detail, how does back-propagation work in this case?
- Why was this approach chosen? Does it promise to be robust or good for low data situations specifically?
- The fonts in Figure 2 are very small, please make them bigger, and the Figure may not print well in b/w. Why does the mean of the absolute weights go up for high buffer positions? Is there some "leaking" from even longer contexts?
- I don't understand "However, human speech is not deterministic and one cannot expect [...] truth". You are saying that the model cannot be excepted to reproduce the input exactly? Or does this apply only to the temporal distribution of the sequence (but not the spectral characteristics)? The previous sentence implies that it does. And how does teacher-forcing help in this case?
- what type of speed is "x5"? Five times slower or faster than real-time?

Experiments
- Table 2: maybe mention how these results were computed, i.e. which systems use optimized hyper parameters, and which don't? How do these results support the interpretation of hte results in the introruction re in-the-wild data and found data?
- I am not sure how to read Figure 4. Maybe it would be easier to plot the different phone sequences against each other and show how the timings are off, i.e. plot the time of the center of panel one vs the time of the center of panel 2 for the corresponding phone, and show how this is different from a straight line. Or maybe plot phones as rectangles that get deformed from square shape as durations get learned?
- Figure 5: maybe provide spectrograms and add pitch contours to better show the effect of the dfifferent intonations?
- Figure 4 uses a lot of space, could be reduced, if needed

Discussion
- I think the first claim is a bit to broad - nowhere is it shown that the method is inherently more robust to clapping and laughs, and variable prosody. The authors will know the relevant data-sets better than I do, maybe they can simply extend the discussion to show that this is what happens.
- Efficiency: I think Wavenet has also gotten much faster and runs in less than real-time now - can you expand that discussion a bit, or maybe give estimates in times of FLOPS required, rather than anecdotal evidence for systems that may or may not be comparable?

Conclusion
- Now the advantage of the proposed model is with the number of parameters, rather than the computation required. Can you clarify? Are your models smaller than competing models?

---

> ### Author Response · Authors · 2017-12-12
> **reply (part 1 of a long reply, due to the very detailed comments)**
>
> We thank AnonReviewer2 for the constructive and thoughtful comments. Our method was designed, and our interests lie, specifically in Speech Synthesis. Applications of our methods to recognition and other tasks are of interest to us but not our focus. Unlike much of the previous work (Char2Wav is an exception) we publish our code and models , thus we allow and encourage a direct comparison with our method. The shortcomings of previous work cannot be held against us.
>
> More generally, neural speech synthesis is an emerging topic in representation learning and generative models, both central to the ICLR community. There is sizable interest within the ICLR community in speech synthesis, which became in the last few years a fast paced field with a constant stream of new results. Char2Wav, SampleRNN, “Fast Generation for Convolutional Autoregressive Models” ICLR'17 are three recent examples, as well as many submissions to ICLR'18.
>
> We are sorry that AnonReviewer2 was not convinced by the link to the phonological loop. We can of course remove this part without taking away nothing from the paper's clarity, technical novelty and experimental success. The link is said explicitly to be an inspiration only.
>
> However, the link to phonological loop fascinates us, and we observe in the model necessities that exist in Baddeley's model.
> 1. By phonological, we don't mean information related necessarily to phonemes, as the review seems to imply. Rather, we mean a joint (mixed) representation, in memory, of sound based information and language based information, which is a unique characteristic of our model.
> 2. The articulacy information in our model does not correspond to a physical model of the human vocal tract. Our model synthesizes a WAV using vocoder features and the network that generates vocoder features is our analog of an articulacy system.
> 3. Other important aspects of the Baddeley's phonological loop include a short term memory, which we have, and a rehearsal mechanism.  The analog to a rehearsal mechanism is the recursive way in which our buffer is updated. Namely, the new element in the buffer u is computed based on the entire buffer. We would like that stress that without this part, our model is completely ineffective, as noted in Sec. 3, description of step III.
>
> The text of the discussion has been updated and it now reads as follows. If this is objectionable, we can remove the entire comparison to Baddeley's work.
> “The link we form to the model of Baddeley is by way of analogy and, to be clear, does not imply that we implement this model as is. Specifically, by phonological features, we mean a joint (mixed) representation, in memory, of sound based information and language based information, which is a unique characteristic of our model in comparison to previous work. The short term memory in Baddleley's model is analog to our buffer and the analog to the rehearsal mechanism is the recursive way in which our buffer is updated. Namely, the new element in the buffer (u) is calculated based on the entire buffer. As noted in Sec. 3, without this dependency on the buffer, our model becomes completely ineffective.“
>
> Following the reviewers' request, we have added more details to Sec. 3.2.
>
> Regarding our model belonging to the family of RNNs, we have addressed this in our reply to AnonReviewer1. We meant to highlight the differences from conventional and other existing RNN models, and this is not clarified.
>
> The link between the model properties and the ability to fit speakers with less and lower-quality data is a hypothesis only (and presented as such). “In the wild” is an appealing application that is enabled by this capability, not the only one. The hypothesis above is based on the commonsense assumption that simplicity leads to robustness. Since our architecture employs a simple reader, a shared memory and shallow networks it is simpler than other architectures. We present this link as a hypothesis and do not test it directly since it is extremely hard to build a hybrid system (that has some of the properties) which works.
>
> On Blizzard 2011, our results are better than Tacotron (reimplementation) but not significantly better than Char2Wav, while on Blizzard 2013 it is significantly better than both. This can be attributed to the clean nature of Blizzard 2011, for which Char2Wav is robust enough and therefore does support the robustness to noise claims. The text of the paper is updated to address this and now reads:
> “It is interesting to note that on Blizzard 2011, our results are better than Tacotron (reimplementation) but not significantly better than Char2Wav, while on Blizzard 2013 it is significantly better than both. This can be attributed to the clean nature of Blizzard 2011, for which Char2Wav is robust enough, and demonstrates our method's robustness to noise.“

---

> > ### Author Response · Authors · 2017-12-12
> > **reply (part 2)**
> >
> > Thank you for pointing us to the missing reference. We have added a new text to the previous work section:
> > “HMM-based methods require careful collection of the samples, or as recently attempted by~Baljekar et al., filtering of noisy samples for in-the-wild application.“
> >
> > Following the reviewer's suggestion, we have computed Mel cepstral distortion (MCD) scores. This is an automatic, albeit very limited, method of testing compatibility between two audio sequences. Since the sequences are not aligned, we employ MCD DTW, which uses dynamic time warping (DTW) to align the sequences. The results below correspond to Tab. 3 and 5 in the current paper (these were numbered 2 and 3 before). As can be seen, our method outperforms the baseline methods in this score as well, except for one single speaker experiment, where Tacotron achieves a lower distortion. As can be seen in the MOS data for the very same experiment, Tacotron is not really performing well in this experiment. These results are now added to the paper as Tab. 4 and 6.
> >
> >
> >                         	LJ	       Blizzard 2011 Blizzard 2013
> > Tacotron	12.82+-1.41	14.60+-7.02	---
> > Char2Wav	19.41+-5.15	13.97+-4.93	18.72+-6.41
> > VoiceLoop	14.42+-1.39	8.86+-1.22	8.67+-1.26
> >
> >                 	VCTK22	VCTK65	VCTK85	VCTK101
> > Char2Wav	15.71+-1.82	15.1+-1.45	15.23+-1.49	15.06+-1.32
> > VoiceLoop	13.74+-0.98	14.1+-0.94	14.16+-0.87	14.22+-0.88
> >
> > Following the reviewer's request, we have added more text to the part of the paper that describes the attention mechanism, which is based on Gaussian Mixture Models. The added text reads:
> > “The loss function of the entire model depends on the attention vector through this context vector. The GMM is differentiable with respect to mean, std and weight, and these are updated, during training, through backpropagation.”
> >
> > As mentioned, the attention mechanism was selected since it's monotonic and since it was successfully employed in a pervious speech synthesis work. We have experimented with slightly modified versions since, which, for example, select the mixture component with the maximal probability instead of a weighted average. This seems to work somewhat better.
> >
> > Following the reviewer's suggestion, Figure 2 has been remade. The phenomena the reviewer noted, that the weights tend to increase at the end suggest that there might be valuable information beyond the memory horizon. However, as noted in  our response to AnonReviewer1, longer buffers did not result in a noticeable improvement .
> >
> > Regarding the sentence on human speech. We simply meant that even the same speaker cannot replicate her voice to completely remove the MSE loss since there is variability that is present between every time a sentence is spoken. Teacher forcing solves this since it eliminates most of the drift and enforces a specific way of uttering the sentence. A clarification has been added to the paper as follows:
> > “For example, even the same speaker cannot replicate her voice to completely remove the MSE loss since there is variability when repeating the same sentence. Teacher forcing solves this since it eliminates most of the drift and enforces a specific way of uttering the sentence.“
> >
> > By 5x we mean 5 times faster than its CPU implementation, which is near real-time. This is now made clear in the paper.
> >
> > For the query on which systems use optimized hyper parameters: the Tacotron reimplementations were optimized by the community to work best on each of the datasets given. For Char2Wav, we made the best effort to find the best hyper parameters for each dataset, including in the wild datasets.
> >
> > Following the review, we have added the following text to the paper:
> > “The training of the Char2Wav model, in each experiment, was optimized by measuring the loss on the validation set, over the following hyperparameters: initial learning rate of [1e-2,1e-3, 1e-4], source noise standard deviation [1,2,4], batch-size [16,32,64] and the length of each training sample [10e2, 10e4]. “
> >
> > Regarding the clarity of Fig. 4, from the description we understand that Fig. 3 is questioned. The suggested visualizations are indeed suitable. However, the figure, as provided, has the advantage of depicting the actual (raw) probabilities. Before submitting the paper, we tried various other ways to visualize, most resulted in “less-scientific” or cluttered plots.
> > Following the review, we added the 4th Mel-cepstrum coefficient for the three speakers. Each is compared against the ground-truth of the first speaker, illustrating further the differences between different speakers.

---

> > > ### Author Response · Authors · 2017-12-12
> > > **reply (part 3)**
> > >
> > > Thank you for the suggestion to Fig. 5. The figure has been updated.
> > > Following the reviewer's suggestion for figure 5, the caption of figure 5 now reads: “Same input, different intonations. A single in the wild speaker saying the sentence ``priming is done like that '', where each time $S_0$ is initialized differently. (a) Without priming. (b) Priming with the word ``I". (c) Priming with the word ``had''. (d) Priming with the word ``must''. (e) Priming with the word ``bye''. The figure shows the raw waveform, spectrogram, and F0 estimation (include voicedness) in the first, second and third rows respectively. From the spectrogram plots we can observe different duration for some phonemes. The F0 estimation of (c) and (d) shows that the speaker talks in higher tone while in (b) and (e) we can observe lower tone of the speaker. This demonstrates how priming changes the intonations of the model outputs.”
> > >
> > > Following the reviewer's suggestion, we have added details to the first paragraph of the discussion. Its last sentence now reads as follows
> > > “As our experiments show, our method is mostly robust to these, since it is able to model the voices despite of these difficulties and without replicating the background noises in the synthesized output. The baseline model of Char2Wav was not able to properly model the voices of the youtube dataset and presented clapping sounds in its output.”
> > >
> > > In light of the importance of the wavenet model to the industry, considerable effort has been invested in speeding up the formidably slow inference-time of it. Most of these engineering efforts focus on eliminating redundancy and on efficient software/hardware utilization. A month after our submission, a new model “parallel wavenet” emerged, which has comparable run time to our (unoptimized) approach. Unlike our model, it is not based on attention (that requires sequential computation) but on linguistic features.
> > >
> > > Regarding the number of parameters, please see our response to AnonReviewer3. The number of parameters is similar to Multispeaker Tacotron, but our architecture is much simpler. It is considerably lower than that of DV2. As shown in the response to AnonReviewer3, we can compress the number of parameters by half and maintain a reasonable performance. The conclusions were slightly altered in order to reflect this.

---

### Official Review · AnonReviewer3 · 2017-11-30
**A good paper, but it could be better for writing and baseline comparisons**

**Rating:** 6
**Confidence:** 4

**Review:**

This paper studies the problem of text-to-speech synthesis (TTS) "in the wild" and proposes to use the shifting buffer memory.

Specifically, an input text is transformed to phoneme encoding and then context vector is created with attention mechanism. With context, speaker ID, previous output, and buffer, the new buffer representation is created with a shallow fully connected neural network and inserted into the buffer memory. Then the output is created by buffer and speaker ID with another fully connected neural network. A novel speaker can be adapted just by fitting it with SGD while fixing all other components.

In experiments, authors try single-speaker TTS and multi-speaker TTS along with speaker identification (ID), and show that the proposed approach outperforms baselines, namely, Tacotron and Char2wav. Finally, they use the challenging Youtube data to train the model and show promising results.

I like the idea in the paper but it has some limitations as described below:

Pros:
1. It uses relatively simple and less number of parameters by using shallow fully-connected neural networks.
2. Using shifting buffer memory looks interesting and novel.
3. The proposed approach outperforms baselines in several tasks, and the ability to fit to a novel speaker is nice. But there are some issues as well (see Cons.)

Cons:
1. Writing is okay but could be improved. Some notations were not clearly described in the text even though it was in the table.
2. Baselines. The paper says Deep Voice 2 (Arik et al., 2017a) is only prior work for multi-speaker TTS. However, it was not compared to. Also for multi-speaker TTS, in (Arik et al., 2017a), Tacotron (Wang et al., 2017) was used as a baseline but in this paper only Char2wav was employed as a baseline. Also for Youtube dataset, it would be great if some baselines were compared with like  (Arik et al., 2017a).


Detailed comment:
1. To demonstrate the efficiency of the proposed model, it would be great to have the numbers of parameters for the proposed model and baseline models.
2. I was not so clear about how to fit a new speaker and adding more detail would be good.
3. Why do you think your model is better than VCTK test split, and even VCTK85 is better than VCTK101?

---

> ### Author Response · Authors · 2017-12-12
> **reply**
>
> Thank you very much for your support and constructive feedback. As you noted, we have attempted to explain the details of the architecture both in a table and in the text. In a few places, details that appeared in the table were omitted from the text. This is now corrected.
>
> Further baselines: reimplementing Deep Voice, which is a well engineered and complex system, is beyond our capabilities, and at the time of submission, beyond the capabilities of the open source community. Tacotron was converted, with a significant effort and partial success, by the authors (Arik et al. 2017a) into a multispeaker system. This, too, required an effort that is beyond our resources; the multi-speaker feature is also absent in the many reimplementations of Tacotron. Despite a considerable effort, we were not able to get Char2Wav to synthesize speech when trained on the YouTube dataset.
>
> To the detailed comments 1--3
> ==========================
> 1. Below is a table describing the number of parameters in each approach.
> Loop:                9.3 * 10^6    (9,332,060)
> Char2Wav:       26.5 * 10^6 (26,494,492)
> DeepVoice 2:   29.6 * 10^6  (29,649,888)
> Multispeaker Tacotron:     9.2 * 10^6 (9,212,636)
>
> This table does not reflect the relative simplicity of our method, since the number of parameters is hindered by fully connected layers and does not reflect, for example, the sophisticated nature of Tacotron's CBHG structures in comparison to our fully connected layers. In addition, we made, by the time of submission, no attempt to minimize the number of parameters. Since then, we were able to replicate our results with hidden layers of size (dk/30) instead of the arbitrary dk/10 we used in the paper. This results in a total number of parameters that is only a half of the number of parameters (5.7M) and in a small loss of performance. The MOS for this smaller model (Loop-bottleneck) are below.
>
> Multi Speaker	  vctk22 model
> Char2wav	          2.78+-1.00
> Loop	                  3.54+-0.96
> GT	                          4.63+-0.63
> Loop - bottleneck	  3.17+-0.98
>
>
> 2. Following the reviewers' request, we have added more details to Sec. 3.2.
>
>
> 3. The Top-1 identification scores are computed by a multi-classification speaker network. Better classification rates with generated samples is expected since the generated distribution lies closer to the training samples distribution than the test distribution. Also, since VCTK85 has only 85 classes, it is expected to perform marginally better than VCTK101, which has 101 classes.

---

### Comment · AnonReviewer1 · 2017-11-22
**a good paper.  accept.**

This is an interesting paper investigating a novel neural TTS strategy that can generate speech signals by sampling voices in the wild.  The main idea here is to use a working memory with a shifting buffer.  I also listened to the samples posted on github and the quality of the generated voices seems to be OK considering that the voices are actually sampled in the wild. Compared to other state-of-the-art systems like wavenet, deep voice and tacotron, the proposed approach here is claimed to be simpler and relatively easy to deploy in practice.   Globally this is a good piece of work with solid performance. However, I have some (minor) concerns.

1.  Although the authors claim that there is no RNNs involved in the architectural design of the system,  it seems to me that  the working memory with a shifting buffer which takes the previous output as one of its inputs is a network with recurrence.

2. Since the working memory is the key in the architectural design of VoiceLoop, it would be helpful to show its behavior under various configurations and their impact to the performance. For instance,  how will  the length of the running buffer affect the final quality of the voice?

3. A new speaker's voice is generated by only providing the speaker's embedding vector to the system.  This will require a large number of speakers in the training data in the first place to get the system learn the spread of speaker embeddings in the latent (embedding) space.  What will happen if a new speaker's acoustic characteristics are obvious far away from the training speakers?  For instance, a girl voice vs. adult male training speakers.  In this case, the embedding of the girl's voice will show up in the sparse region of the embedding space of training speakers.  How does it affect the performance of the system?  It would be interesting to know.

---

> ### Public Comment · (anonymous) · 2017-11-23
> **reply**
>
> Thank you for your constructive comments.
>
> 1. The proposed memory model is indeed a network with recurrence and, therefore, it is a Recurrent Neural Network. We meant to highlight the differences from conventional and other existing RNN models. We will clarify this point.
>
> 2. In our experiments, buffer sizes between 15 and 30 seem to produce similar results. Less than 15 is detrimental. See also Fig. 2, which depicts the relative contribution of each column of the buffer and shows that all columns impact the computations done for the attention, the output, and the buffer update.
>
> 3. We have not tested the specific experiment that is described but have done a similar experiment in which we train on voices of north Americans and fit on other accents. The results confirm that when fitting out of distribution voices, the quality degrades. However, the fitting is still successful in capturing the pitch and other aspects of the voice.
>
> The authors.

---

### Decision · Program_Chairs · 2018-01-29
**ICLR 2018 Conference Acceptance Decision**

**Decision:**

Accept (Poster)

**Comment:**

Meta score: 7

This paper presents a novel architecture for neural network based TTS using a memory buffer architecture.  The authors have made good efforts to evaluate this system against other state-of-the-art neural TTS systems, although this is hampered by the need for re-implementation and the evident lack of optimal hyperparameters for e.g. tacotron.  TTS is hard to evaluate against existing approaches, since it requires subjective user evaluation.  But overall, despite its limtations, this is a good and interesting paper which I would like to see accepted

Pros:
 - novel architecture
 - good experimentation on multiple databases
 - good response to reviewer comments
 - good results

Cons:
 - some problems with the experimental comparison (baselines compared against)
 - writing could be clearer, and sometimes it feels like the authors are slightly overclaiming

I take  the point that this might be more suitable for a speech conference, but it seems to me that paper offers enough to the ICLR community for it to be worth accepting.